# A Structure Variation in *qPH8.2* Detrimentally Affects Plant Architecture and Yield in Rice

**DOI:** 10.3390/plants12183336

**Published:** 2023-09-21

**Authors:** Wenqiang Sun, Qiang Sun, Li Tian, Yongjian Sun, Sibin Yu

**Affiliations:** 1National Key Laboratory of Crop Genetic Improvement, Huazhong Agricultural University, Wuhan 430070, China; wqsun@webmail.hzau.edu.cn (W.S.); sunyongjian28576@163.com (Y.S.); 2College of Life Science and Technology, Huazhong Agricultural University, Wuhan 430070, China; 3College of Plant Science and Technology, Huazhong Agricultural University, Wuhan 430070, China; sq_sunqiang1@163.com (Q.S.); litian_ccc@163.com (L.T.)

**Keywords:** rice, chromosomal segment substitution line, plant height, QTL, structure variation

## Abstract

Plant height is an important agronomic trait associated with plant architecture and grain yield in rice (*Oryza sativa* L.). In this study, we report the identification of quantitative trait loci (QTL) for plant height using a chromosomal segment substitution line (CSSL) population with substituted segments from *japonica* variety Nipponbare (NIP) in the background of the *indica* variety 9311. Eight stable QTLs for plant height were identified in three environments. Among them, six loci were co-localized with known genes such as *semidwarf-1* (*sd1*) and *Grain Number per Panicle1* (*GNP1*) involved in gibberellin biosynthesis. A minor QTL *qPH8.2* on chromosome 8 was verified and fine-mapped to a 74 kb region. Sequence comparison of the genomic region revealed the presence/absence of a 42 kb insertion between NIP and 9311. This insertion occurred predominantly in temperate *japonica* rice. Comparisons on the near-isogenic lines showed that the *qPH8.2* allele from NIP exhibits pleiotropic effects on plant growth, including reduced plant height, leaf length, photosynthetic capacity, delayed heading date, decreased yield, and increased tiller angle. These results indicate that *qPH8.2* from temperate *japonica* triggers adverse effects on plant growth and yield when introduced into the *indica* rice, highlighting the importance of the inter-subspecies crossing breeding programs.

## 1. Introduction

Rice (*Oryza sativa* L.) is one of the most important crops, providing a staple food for half of the world’s population [1]. Plant height, controlled by the number of elongated internodes and the length of internodes, is a determinant trait owing to its essential roles in plant architecture and grain yield. Proper plant height improves lodging resistance, fertilizer absorption efficiency, and grain yield in crops [2]. Thus, understanding the genetic mechanisms underlying plant height is crucial for crop improvement.

Quantitative trait loci (QTL) mapping studies have confirmed that plant height is a complex trait controlled by several major and minor loci and affected by kinds of plant hormones and external environmental factors [3]. More than one hundred genes related to plant height have been identified and distributed on the 12 chromosomes in rice (http://archive.gramene.org/qtl/, accessed on 1 September 2023). Many plant height genes are related to the biosynthesis and regulation of phytohormones, such as gibberellin acid (GA) and auxin [2,4]. The semidwarf-1 (*sd1*) gene is the ‘green revolution gene’ that significantly boosts crop yield by increasing the harvest index and reducing the probability of rice lodging before harvest [5]. *SD1* encodes GA20 oxidase-2 (GA20ox-2) to catalyze late steps of the active GA biosynthesis [6]. The loss-of-function mutation of GA20ox-2 results in dwarf or semi-dwarf plants. Several mutated alleles of *SD1* have been discovered in germplasm and are being widely used in rice breeding [7]. *Grain Number per Panicle1* (*GNP1*) encodes GA20ox-1 participating in the GA biosynthesis and was proven to affect rice plant height and grain number [8,9]. It is worth noting that most of the plant height genes play a pleiotropic effect on plant architecture and yield-related traits. For example, *NARROW LEAF1* (*NAL1*) encoding a serine protease controls flag leaf width, panicle size, and panicle number and influences plant height by affecting polar auxin transport [10,11,12,13]. The transcription factors *Ghd7* and *Ghd8* regulate plant height, heading date, and grain number by promoting cell division [14,15]. These findings reveal complex and diverse genetic mechanisms in regulating plant height in rice.

The chromosome segment substitution line (CSSL) is a powerful tool for QTL identification for complex traits in plants. Remarkably, the CSSL shows a great advantage in identifying minor-effect QTLs as it contains only one or a few introduced segments in the same background, which can eliminate background interference. In addition, the CSSL provides a good foundation for the quick development of near-isogenic lines (NIL) and secondary segregating populations for genetic validation and fine-mapping of the interested QTLs [16,17]. *Indica* and *japonica* rice are the two major subspecies of cultivated rice. Extensive genetic variations, including single nucleotide polymorphisms and structure variations (SV) between these subspecies, are associated with phenotypic variations in plant growth and development and environmental adaptability [18,19,20,21]. Previously, a set of CSSLs composed of 122 lines, each carrying a particular introduced segment from the *japonica* variety Nipponbare (NIP) in the background of *indica* variety 9311, was developed. We identified many QTLs related to seed dormancy and hybrid vigor using this CSSL population [22,23].

In this study, using the NIP/9311 CSSL population, we identified eight stable QTLs for plant height in three environments. Six loci were co-localized with some reported genes (*SD1*, *GNP1*, and *NAL1*). A novel minor QTL (*qPH8.2*) was identified and delimited to a 74 kb region carrying several candidate genes for plant architecture. Analysis of the NILs containing the presence and absence of this region revealed that the *qPH8.2* alleles from NIP have pleiotropic effects on plant growth with reduced plant height, leaf length, photosynthetic capacity, and grain yield, delayed heading date, and increased tiller angle in the *indica* background.

## 2. Results

### 2.1. QTLs Detected for Plant Height in the CSSL Population

We used a set of CSSLs with introduced chromosome segments from NIP in the 9311 background to survey plant height variation under three environments. NIP is a semi-dwarf variety with a plant height of 73 cm, while 9311 is a moderately taller variety with a plant height of 118 cm (Figure 1a). CSSLs showed a considerable variation in plant height, ranging from 81 to 175 cm, with a mean of 117 cm, and exhibited a quantitative trait inheritance pattern. Most of these lines displayed plant height similar to 9311, but several lines had significantly higher or lower values than 9311 (Figure 1b,c). The lines with extreme higher or lower values may contain genes associated with plant height in the introduced NIP segments. A ridge regression analysis of the CSSL population with 387 bin markers identified twelve QTLs for plant height at Hainan in 2016 (E1), nine at Wuhan in 2016 (E2), and eight at Wuhan in 2017 (E3), respectively. Among these QTLs, seven QTLs on chromosomes 1, 3, 4, and 8 were detected repeatedly in three environments, and one QTL (*qPH8.1*) revealed a significant effect (*p* < 0.01) at E2 and E3, but marginal significance (*p* = 0.051) at E3 (Table 1). These eight stable QTLs explained approximately 80% of phenotypic variance in the population, and half of them with NIP alleles had a negative effect on plant height. Of them, *qPH1.2* and *qPH1.3*, explaining more than 10% of the phenotypic variances, were two major QTLs. *qPH1.3*, located at the *semidwarf-1* gene *sd1*, had the largest additive effect, with the NIP allele increasing height and explaining approximately 37.7% of the phenotypic variance. The other five of the eight loci were co-localized with several corresponding genes associated with plant height, i.e., *OsGA2ox3*, *GNP1*, *SSD1*, *NAL1*, and *Ghd8* (Figure 1d; Table 1). Notably, *qPH8.2*, a minor QTL on chromosome 8 that negatively affects plant height, was not reported before.

### 2.2. Genetic Validation of qPH8.2

To evaluate the genetic effect of *qPH8.2*, the line (CSSL32) that contains an introduced NIP segment covering *qPH8.2* was selected for genetic validation (Figure 2a,b). The plant height of CSSL32 was 26.1 cm lower than that of 9311. On the other side, CSSL32 showed an increase in heading date by 5.2 days and tiller angle by 15.5° compared to 9311 (Figure 2c–e).

The graphic genotype of CSSL32 showed three other introduced NIP segments on chromosomes 1, 3, and 10 (Figure 2b). To verify the genetic effect of *qPH8.2*, CSSL32 was crossed with 9311 to generate a CSSL-derived F_2_ population. Then, a small F_2_ population composed of 288 individuals was genotyped by using six polymorphic markers (01C131, 03C041, RM5556, M9060, M1508, and 10C051) to target the four introduced segments (Figure 2b). ANOVA of three genotypes at each marker revealed that only the two markers, M9060 and M1508, located in the *qPH8.2* region on chromosome 8, were significantly associated with plant height (Appendix A). A segregation ratio of approximately 1:2:1 for plant height phenotype suggested that the short phenotype was caused by a recessive gene at a single locus. The individuals with homozygous NIP genotype at M9060 showed shorter plant height, later heading date, and larger tiller angle than those with homozygous 9311 genotype or heterozygous genotype in this F_2_ population (Figure 2f–h). These results validated the effect of *qPH8.2* in regulating plant height and revealed that the NIP allele is recessive to the 9311 allele at *qPH8.2*. Together with seven additional polymorphic markers across *qPH8.2*, a linkage map is constructed (Appendix A). Further, QTL analysis of mapped *qPH8.2* to the interval between the markers M8061 and M1077 was conducted. *qPH8.2* explained 60.3% of the phenotypic variance of plant height, 50.8% of the phenotypic variance of heading date, and 19.2% of the tiller angle variance in the F_2_ population (Figure 3a). These results indicate that *qPH8.2* is a pleiotropic QTL underlying plant height, heading date, and tiller angle.

### 2.3. Fine Mapping of qPH8.2

To perform fine mapping of *qPH8.2*, a large CSSL-derived segregating population comprising 4200 individuals was used to select the recombinants between the markers M8061 and M1077. Eight informative recombinants were obtained using eight additional markers (Appendix A, Appendix A). A progeny test of plant height, heading date, and tiller angle for the informative recombinants delimited *qPH8.2* into a 74 kb region flanked by markers M9211 and M9285 (Figure 3b).

Sequence comparison of NIP and 9311 in this 74 kb region revealed the presence of a 42 kb insertion in NIP but an absence of the insertion in 9311 (Figure 3c). Two predicted genes (*Loc_Os08g15220* and *Loc_Os08g15230*) are outside the 42 kb region. *Loc_Os08g15220* encodes a retrotransposon. *Loc_Os08g15230* encodes a heat shock protein. However, the sequence comparison of *Loc_Os08g15230* showed no nucleotide variation between NIP and 9311. The 42 kb insertion may be the putative region harboring the causal gene for plant height. In this 42 kb insertion region, there are 20 predicted genes based on the reference NIP genome. According to the expression profile database (http://rice.uga.edu/, accessed on 5 April 2023), 10 of 20 genes were expressed—in at least one tissue, such as shoot, leaf, or inflorescence (Figure 3d). Examining the expression of these genes, we found that only *Loc_Os08g15296* was expressed in the shoot, leaf, and inflorescence, whereas the other genes, except for *Loc_Os08g15312*, were hardly detectable in the shoot. Therefore, *Loc_Os08g15296* encoding a photosystem II reaction center protein H may be the promising gene for *qPH8.2*.

### 2.4. Frequency of the 42 kb Insertion in Rice Germplasm

To analyze the presence or absence of the 42 kb insertion in rice accessions, we downloaded the sequences of the 74 kb region in 86 rice accessions, which include ten temperate *japonica* rice, fifteen tropic *japonica* rice, thirty-nine *indica* rice, eight *aus* rice, seven basmati rice, and seven wild relatives (http://ricerc.sicau.edu.cn/, accessed on 6 July 2023). In the *japonica* subgroup, sequence alignment analyses found that the 42 kb insertion occurred in all 10 surveyed temperate *japonica* accessions, but the absence of the insertion was observed in the 15 tropic *japonica* accessions, as well as in the 39 *indica* rice and other remaining accessions (Appendix A). These results indicate that the 42 kb insertion is predominantly present in temperate *japonica* rice.

### 2.5. Pleiotropic Effects of qPH8.2 on Photosynthetic Characteristics and Plant Yield

To evaluate the genetic effect of *qPH8.2* precisely, a pair of near-isogenic lines (NILs), NIL-NIP and NIL-9311, were developed, which carry only a 1.3-Mb segment covering the contrasting NIP and 9311 alleles at *qPH8.2* within the 9311 background. Compared to NIL-9311, NIL-NIP showed a decrease in plant height by 20.1 cm, a delay in heading date by 4.5 days, and an increase in tiller angle by 16° at the heading stage, which was consistent with the effect of *qPH8.2* in CSSL32. In addition, NIL-NIP also decreased plant height by 29.4% and 34.7% at the seedling and tillering stages, respectively (Figure 4a–n). Following the decrease in plant height, NIL-NIP significantly reduced the internode length and the panicle length (Figure 4d,e,j,k). In addition, NIL-NIP shortened the flag leaf length by 26% (Figure 4f). Notably, except for reducing plant height, *qPH8.2* from NIP also adversely affected panicle number, grain number per panicle, 1000-grain weight, and grain yield per plant (Figure 4o–r). These results indicate that *qPH8.2* inhibits plant growth and grain yield of rice.

As two candidate genes were involved in photosynthesis, the photosynthetic parameters of the flag leaves at the heading stage were measured on the two NILs to confirm whether *qPH8.2* affects photosynthetic capacity. As shown in Figure 4, compared to NIL-9311, the photosynthetic rate was 12.2% lower in NIL-NIP, and NIL-NIP also reduced the stomatal conductance, transpiration rate, and intercellular CO_2_ concentration (Figure 4s–v). These results suggest that *qPH8.2* may inhibit plant growth by repressing photosynthetic capacity.

## 3. Discussion

CSSLs, as a permanent genetic resource, allow phenotypic evaluation repeatedly under various environments, making a significant advantage in verifying the stability of QTLs of interest. In the present study, we investigated phenotypic variation in the NIP/9311 CSSL population under three environments and identified eight stable QTLs for plant height. Among them, six loci co-localized in the same or overlapping regions of known genes control plant height (Figure 1d; Table 1). For example, *qPH1.2*, *qPH1.3*, and *qPH3.2* co-localized with *OsGA2ox3*, *SD1*, and *GNP1*, respectively, which all influence plant height by regulating active gibberellin levels [6,9,24]. *qPH3.1* was mapped to a 200 kb region that contains the reported gene *SSD1*, which determines organ elongation by regulating cell division [25]. *qPH4* overlapped with the gene *NAL1*, which affects plant height by regulating polar auxin transport [10,11,12,13]. *qPH8.1* was located near *Ghd8*, which exhibits a pleiotropic effect on plant height, heading date, and yield [15]. Interestingly, *Ghd8* plays an opposite function controlling heading date, delayed flowering under the long-day condition, but promoted it under the short-day condition. It may explain that *qPH8.1* was detected at E2 and E3 both under long-day conditions but did not reach a significant level (*p* = 0.01) at E2, a short-day condition. These results indicate that *qPH8.1* (*Ghd8*) may regulate plant height depending on the environment.

As it contains one or a few specifically introduced segments, the CSSL is an efficient start-up to quickly generate CSSL-derived populations and NILs by a backcrossing scheme to validate and finely map QTLs of interest [26]. In the present study, we detected a minor-effect QTL, *qPH8.2*, explaining approximately 6% of the phenotypic variance of plant height in the CSSL population and validated it as a major-effect locus that explained 60% of phenotypic variance in the CSSL-derived F_2_ population. The investigation of NILs revealed a pleiotropic effect of *qPH8.2* on various agronomic traits, including plant architecture, heading date, and yield (Figure 3 and Figure 4; Table 1). These data support that the CSSL population provides a powerful tool for identifying minor QTLs.

One of the notable findings in this study is that the NIP alleles at *qPH8.2* repressed plant growth and yield-related traits. We observed a significant decrease in plant height, leaf length, and panicle length in NIL-NIP compared to NIP-9311 at the seedling and tillering stages (Figure 4). Furthermore, we delimited *qPH8.2* into a 42 kb region. According to the expression profile, the 42 kb insertion region harbors ten expressed genes (Figure 4). Only *Loc_Os08g15296* is expressed at the shoot, leaf, and panicle, which aligns with the pleiotropic effect of *qPH8.2* mentioned above (Figure 3). Therefore, *Loc_Os08g15296* encoding a reaction center protein H, one of the components of the core complex of photosystem II for water-oxidizing and O_2_-evolving in photosynthesis [27], is the most likely candidate for *qPH8.2*. Previous studies revealed that the defect in photosynthetic capacity could repress plant growth [28]. We suppose that this photosystem-related gene in the *indica* background may interfere with photosynthesis. A comparison of the NILs revealed a significant decrease in photosynthetic rate, stomatal conductance, transpiration rate, and intercellular CO_2_ concentration in NIL-NIP compared with NIL-9311 (Figure 4), which provides a piece of supporting data for the hypothesis. However, further transgenic experiments are required to confirm the function of the candidate gene(s).

Many studies have reported that genetic variations, including single nucleotide polymorphisms and SV, are associated with the heritable phenotypic diversity observed within and between species [29]. Understanding the contribution of SV, such as insertion, deletion, duplication, and translocation, to plant phenotypic variation is vital for plant breeders to improve varieties. In rice, a few SV in genes were reported underlying many agronomic traits, such as submergence tolerance [30], phosphorus-deficiency tolerance [31], and plant architecture [32]. However, the function of most SVs between the two main subspecies in rice remains to be determined [19,21]. In the present study, we used the CSSL population that was derived from the *japonica* NIP and the *indica* 9311, two representative genome-sequenced cultivars, to identify QTLs for plant height. Further sequence comparison of NIP and 9311 revealed a 42 kb insertion/deletion SV surrounding *qPH8.2* that impacts plant growth. Notably, when analyzing more genome-sequenced accessions, we found that this 42 kb insertion occurred predominantly in temperate *japonica* but not in *indica*. In addition, the 42 kb segment is absent in wild rice (*O. rufipogon*) (Appendix A), suggesting that it emerged after the differentiation of subspecies and in the domestication process. In breeding programs, many segments along with functional genes introduced from *japonica* into *indica* varieties, or vice versa, occur frequently and may facilitate the breeding selection of desirable agronomic traits. Therefore, understanding the SV region such as *qPH8.2* that underlies multiple agronomic characters will provide insight into the SV function in plant growth and guide the inter-subspecies introgression breeding in rice.

## 4. Materials and Methods

### 4.1. Plant Materials

A CSSL population consisting of 122 lines was developed by marker-aided backcrossing approach, in which the sequenced *japonica* cv. Nipponbare as donor parent and the sequenced *indica* cv. 9311 as a recurrent parent was crossed [22]. The genotypes of the CSSLs were re-analyzed using an SNP chip RICE6K (Illumina) [33]. A bin map based on the recombination breakpoints in the CSSLs was constructed. A total of 387 bins were generated with a median length of 800 kb [22]. The CSSL population was grown in three environments: Hainan, 2016 (E1); Wuhan, 2016 (E2); and Wuhan, 2017 (E3). Each line was planted in a three-row plot with ten plants per row.

To validate the effect of *qPH8.2*, a CSSL line, CSSL32 was selected to backcross with 9311 to generate a CSSL-derived F_2_ population for QTL validation. For fine-mapping of *qPH8.2*, F_2_ individuals heterozygous at the QTL region were self-pollinated to produce a large segregating population. The recombinant individuals were selected using the markers M8061 and M1077 that flank the target QTL. The near-isogenic lines (NILs) carrying alleles at *qPH8.2* within the 9311 background were also developed using a marker-aided backcross scheme. The F_2_ population and NILs were grown at an experimental field at Huazhong Agricultural University, Wuhan, China (30.4° N, 114.2° E). The NILs and progeny of the informative recombinants were grown in a three-row plot with ten plants per row in three replications. Crop management and controlling insects and diseases were carried out following the standard procedures.

### 4.2. Trait Measurement

Several agronomic traits were evaluated in eight individuals of each CSSL. Plant height was measured by the distance from the soil surface to the top of the main panicle (not including the length of the awn). The heading date was measured by the duration from the sowing to the appearance of the main panicle [34]. A protractor (Sanliang, model 187-101, Guangzhou, China) was employed to measure the angle between the most distant tillers on the two sides of the culm base, and half of the angle was treated as the tiller angle of the plant [35]. At the harvest stage, panicle length, panicle number, grain number per panicle, 1000-grain weight, seed setting ratio, and yield per plant were evaluated following the methods described previously [34].

Photosynthetic parameters were measured as described previously [36]. The fully expanded flag leaves on the main stem of each NIL at the heading stage were measured using a portable photosynthesis system (LI-6400XT, LI-COR Biosciences, Lincoln, NE, USA) between am 9:00 and pm 12:30 on clear days. The light level was 1500 μmol m^−2^ s^−1^ provided by red/blue light-emitting diodes. The CO_2_ concentration of the air entering the leaf chamber and the temperature were adjusted to 400 μmol mol^−1^ and 30 °C, respectively. The flow rate to the sample cell was 500 μmol s^−1^. Gas-exchange parameters were recorded once the topmost expanded leaf was enclosed in the chamber and the photosynthesis rate reached a steady state.

### 4.3. DNA Extraction and Genotype Analysis

Total genomic DNA was extracted from fresh, healthy leaves using the CTAB method [37]. The insertion/deletion (Indel) markers (Appendix A) were developed according to the sequence variation between NIP and 9311 (http://ricevarmap.ncpgr.cn, accessed on 4 July 2020) [38]. All Indel and SSR markers were separated using 4% polyacrylamide gel electrophoresis and silver stains for visualization [39].

### 4.4. QTL Mapping

QTL analyses of the normalized data with the bins as markers were performed using a linear ridge regression in the R package “ridge” (http://www.r-project.org/, accessed on 10 September 2018) as described previously [40]. A *t*-test for the ridge regression coefficients was conducted for each bin, which was taken as an independent variable in the linear ridge regression model. A significance level of *p* < 0.01 was set as the threshold to declare the presence of a putative QTL in a given bin. If several adjacent bins showed significant *p* values, then the QTL was tentatively located in the most significant bin with the lowest *p* value. The variance explained by each QTL (bin) was decomposed by using “lmg” from R with the package “relaimpo” [41]. QTL nomenclature followed the principles suggested by McCouch [42].

QTL analysis in the F_2_ population was performed using QTL IciMapping V4.1 (http://www.isbreeding.net/software/, accessed on 10 May 2019) [43]. Means and standard deviation were calculated in Microsoft Excel^®^ 2013 software, and one-way ANOVA comparisons were made to determine significant differences.

### 4.5. The Insertion Analysis in Rice Germplasm

The sequences of the 74 kb region in 86 rice accessions were downloaded from the database of the Rice Resource Center (http://ricerc.sicau.edu.cn/RiceRC/tools/, accessed on 6 July 2023). The sequences were mapped to the NIP genome using minimap2 (v2.26) with parameters of -ax asm20, and the output sam file was converted to a bam file using samtools (v1.14) view. Then, the bam file was sorted and indexed by using samtools sort and samtools index, respectively [44,45].

## 5. Conclusions

CSSLs have a distinct advantage in genetic dissection for complex quantitative traits in plants. In the present study, we conducted the QTL identification for plant architecture in the NIP/9311 CSSL population and detected eight stable QTLs for plant height across three environments, which explained about 80% of phenotypic variance. Among them, a novel minor-effect QTL (*qPH8.2*) was validated as a Mendelian locus and finely mapped to a 74 kb region using a CSSL-derived segregating population. Furthermore, this study found a structure variation with the presence/absence of a 42 kb segment containing *qPH8.2* between *indica* and *japonica*. This segment occurred predominantly in the temperate *japonica* and adversely affected the plant architecture, yield, and photosynthetic capacity if introduced into the *indica* varieties. These findings provide insight into the contribution of the structure variation to the phenotypic variation between *japonica* and *indica* and offer some resources for evaluating and utilizing the genomic introgression between the two subspecies in rice breeding programs.

## Figures and Tables

**Figure 1 plants-12-03336-f001:**
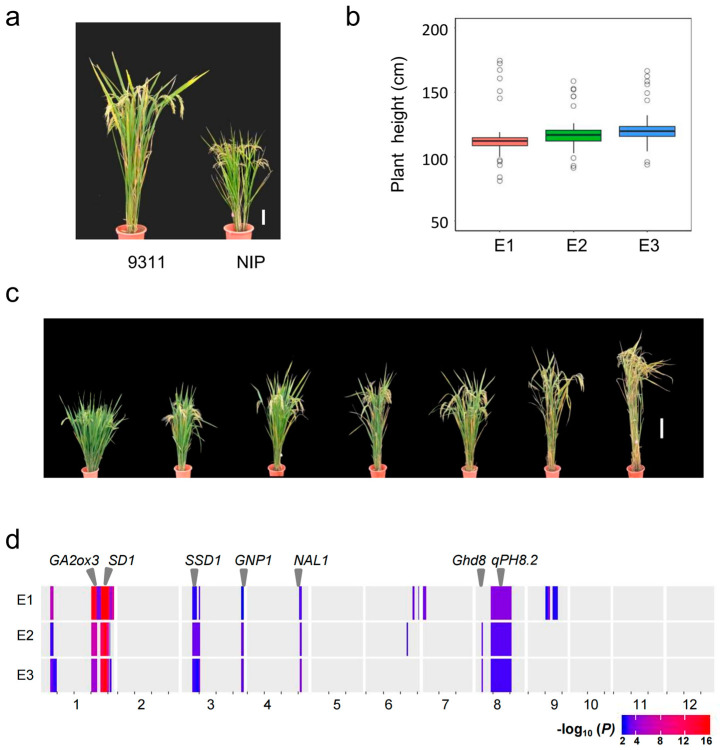
Overview of the QTLs detected for plant height in the NIP/9311 CSSL population. (**a**) Difference in plant height between NIP and 9311 (Scale bar, 10 cm); (**b**) Boxplot of plant height in the CSSLs under three environments. Box edges represent the range of the 25th to the 75th percentiles, with the median value shown by the bold middle line; (**c**) The phenotypic variation of plant height in the CSSL population; (**d**) The relative physical position of QTL detected on the rice chromosome. Some reported genes for plant height are indicated by the triangle on top.

**Figure 2 plants-12-03336-f002:**
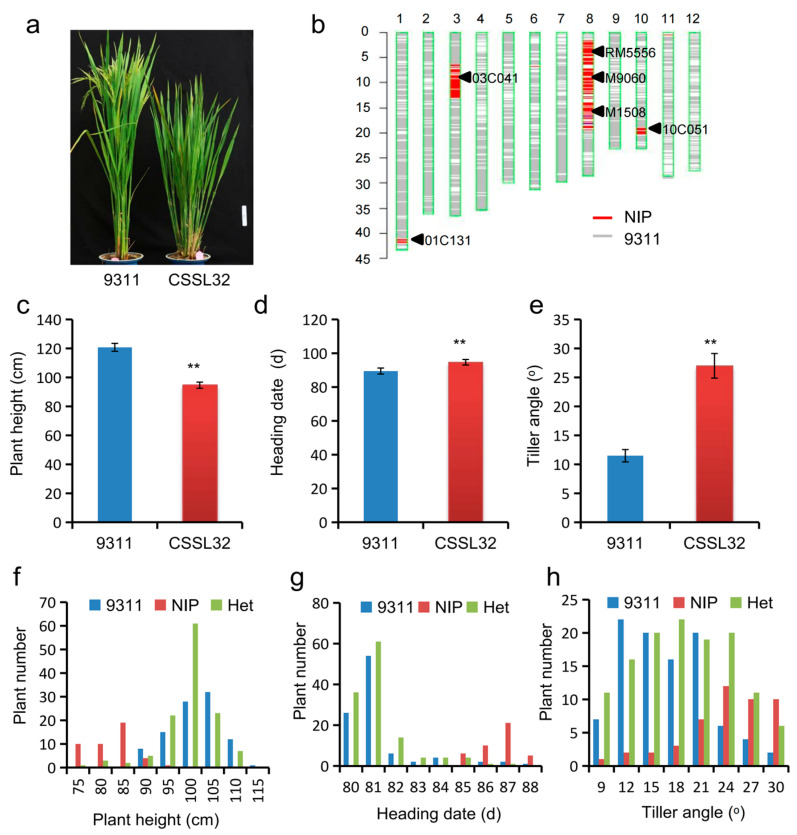
Validation of *qPH8.2* effect. (**a**) Phenotype differences between 9311 (**left**) and CSSL32 (**right**) (Scale bar, 10 cm). (**b**) Graphical genotype of CSSL32 showing the NIP segment at *qPH8.2* and three other segments within the 9311 background. (**c**–**e**) Comparisons of plant height (**c**), heading date (**d**), and tiller angle (**e**). Data are given as mean ± SD (*n* = 20). ** denotes significant differences at *p* < 0.01 by *t*-test; (**f**–**h**) Frequency distribution of plant height (**f**), heading date (**g**), and tiller angle (**h**) of three genotypes assessed by the marker M9060 tightly linked to *qPH8.2*. 9311, NIP, and Het indicate homozygous 9311, NIP, and heterozygous genotypes at the marker M9060, respectively.

**Figure 3 plants-12-03336-f003:**
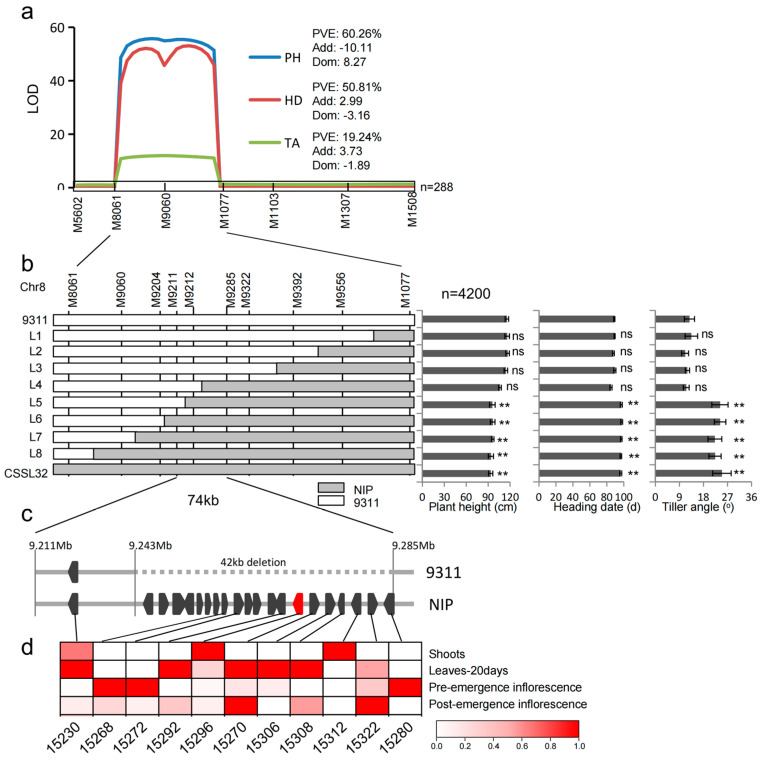
Fine-mapping of *qPH8.2*. (**a**) Primary mapping of *qPH8.2* in a CSSL-derived F_2_ population (*n* = 288) for plant height (PH), heading date (HD), and tiller angle (TA). (**b**) Fine-mapping of *qPH8.2* to a 74 kb interval flanked by the markers M9211 and M9285 using a large population (*n* = 4200). The overlapped recombinant lines are provided with phenotypes by progeny testing. Values are given as mean ± SD (*n* = 8). ** denotes significant differences with 9311 at *p* < 0.01 by *t*-test. (**c**) The insertion region contains some promising genes. (**d**) The expression profile of some candidate genes from the database (http://rice.uga.edu/, accessed on 5 April 2023).

**Figure 4 plants-12-03336-f004:**
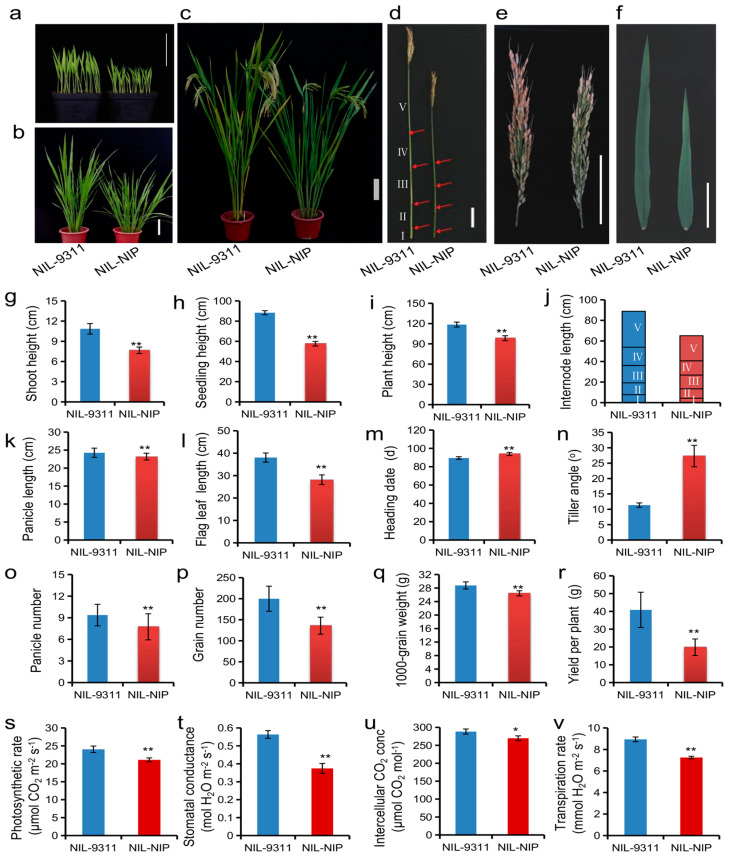
Comparison of agronomic traits and photosynthetic characteristics between NILs of *qPH8.2*. (**a**–**f**) Morphology difference between NIL-9311 and NIL-NIP at the seedling stage with 10 days after germination (**a**), at the tillering stage with 50 days after germination (**b**), and at the heading stage (**c**–**f**). Scale bar: 10 cm. (**g**–**r**) Differences in plant height at the seedling (**g**), tillering (**h**), and heading stage (**i**), internode length (**j**), panicle length (**k**), flag leaf length (**l**), heading date (**m**), and tiller angle (**n**), panicle number (**o**), grain number per panicle (**p**), 1000-grain weight (**q**) and yield per plant (**r**). Data are given as mean ± SD (*n* = 20). ** denotes significant differences at *p* < 0.01 by *t*-test. (**s**–**v**) Differences in photosynthetic rate (**s**), stomatal conductance (**t**), intercellular CO_2_ concentration (**u**), and transpiration rate (**v**). Data are given as Mean ± SD (*n* = 3). * and ** denote significant differences at *p* < 0.05 and *p* < 0.01 by *t*-test, respectively.

**Table 1 plants-12-03336-t001:** The QTLs identified for plant height in the NIP/9311 CSSL population.

Environment ^a^	QTL	Chr	Interval (Mb)	Effect	*p* Value	PVE (%)	Gene
E1	*qPH1.1*	1	6.1–7.6	0.10	1.15 × 10^−6^	6.3	
	*qPH1.2*	1	30.3–33.3	0.15	2.00 × 10^−15^	16.6	*GA2ox3*
	*qPH1.3*	1	38.1–39	0.20	2.00 × 10^−16^	41.1	*SD1*
	*qPH3.1*	3	10.3–10.5	−0.06	2.37 × 10^−3^	3.2	*SSD1*
	*qPH3.2*	3	34.9–end	0.06	9.06 × 10^−3^	2.1	*GNP1*
	*qPH4*	4	31.4–31.6	−0.06	4.01 × 10^−4^	3.6	*NAL1*
	*qPH8.1*	8	3.5–4	−0.03	5.11 × 10^−2^	0.7	*Ghd8*
	*qPH8.2*	8	9.1–20.6	−0.07	1.46 × 10^−4^	5.8	
E2	*qPH1.1*	1	6.1–7.6	0.59	4.44 × 10^−3^	5.6	
	*qPH1.2*	1	30.3–33.3	0.56	1.65 × 10^−7^	11.2	*GA2ox3*
	*qPH1.3*	1	38.1–39	1.23	2.00 × 10^−16^	35.2	*SD1*
	*qPH3.1*	3	10.3–10.5	−0.37	6.75 × 10^−4^	5.7	*SSD1*
	*qPH3.2*	3	34.9–end	0.65	1.25 × 10^−3^	1.7	*GNP1*
	*qPH4*	4	31.4–31.6	−0.26	8.31 × 10^−4^	3.0	*NAL1*
	*qPH8.1*	8	3.5–4	−0.27	1.91 × 10^−3^	2.3	*Ghd8*
	*qPH8.2*	8	9.1–20.6	−0.42	1.78 × 10^−3^	6.2	
E3	*qPH1.1*	1	6.1–7.6	0.70	5.16 × 10^−4^	6.1	
	*qPH1.2*	1	30.3–33.3	0.44	2.32 × 10^−5^	10.2	*GA2ox3*
	*qPH1.3*	1	38.1–39	1.17	2.00 × 10^−16^	36.9	*SD1*
	*qPH3.1*	3	10.3–10.5	−0.42	9.28 × 10^−5^	5.2	*SSD1*
	*qPH3.2*	3	34.9–end	0.62	1.40 × 10^−3^	1.8	*GNP1*
	*qPH4*	4	31.4–31.6	−0.28	1.75 × 10^−4^	5.2	*NAL1*
	*qPH8.1*	8	3.5–4	−0.27	1.36 × 10^−3^	1.8	*Ghd8*
	*qPH8.2*	8	9.1–20.6	−0.38	3.34 × 10^−3^	5.9	

^a^ E1, Hainan, 2016; E2, Wuhan, 2016; E3, Wuhan, 2017.

## Data Availability

Data is contained within the article and Appendix A.

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
