# Peer review of "A Structure Variation in *qPH8.2* Detrimentally Affects Plant Architecture and Yield in Rice"

_plants, 2023, doi:10.3390/plants12183336_

Round 1

Reviewer 1 Report

The study was focused on a structure variation in qPH8.2 detrimentally affects plant architecture and yield in rice. The Authors performed identification of quantitative trait loci (QTL) for plant height using a chromosomal segment substitution line (CSSL) population with substituted segments from japonica variety Nipponbare (NIP) in the background of the indica variety 9311. Eight stable QTLs for plant height were identified in three environments. Among them, six loci were co-localized with the known genes like semidwarf-1 (sd1) and Grain Number per Panicle1 (GNP1) involved in gibberellin biosynthesis. A minor QTL qPH8.2 on chromosome 8 was verified and fine-mapped to a 74-kb region. Comparisons on the near-isogenic lines showed that the qPH8.2 allele from NIP exhibits pleiotropic effects on plant growth, including reduced plant height, leaf length, photosynthetic capacity, delayed heading date, decreased yield, and increased tiller angle.

In my opinion, the paper is quite interesting and valuable, however, I recommend some significant improvements:

-        The discussion should be significantly expanded and deepened to conduct a more complete interpretation of the research results obtained.

-        I suggest supplementing the discussion with available new citations within the research study field.

-        Conclusions should be rewritten, into a more general, as they are currently a rewriting of research results.

-        Authors stated that all Indel and SSR markers were separated using 4% polyacrylamide gel electrophoresis and silver stains for visualization. The electropherograms should be added into the Supplementary file.

-        Extensive editing of English language is required.

 Extensive editing of English language is required.

Author Response

1. The discussion should be significantly expanded and deepened to conduct a more complete interpretation of the research results obtained.

Re: Thanks for your valuable comments. We have rewritten the discussion part (Lines 248-312), highlighting the following aspects: 1) a reason of the unstable qPH8.1 across three environments; 2) the advantage of CSSLs in identification and validation of minor-effect QTL; 3) the possible candidate gene for qPH8.2; 4) the structure variation associated with phenotypic variation in rice.

2. I suggest supplementing the discussion with available new citations within the research study field.

Re: We have checked the literature carefully and cited nine more related references (Ref 13, 19, 26, 27, 29-33) in the discussion and introduction parts in the revised manuscript. 

3. Conclusions should be rewritten, into a more general, as they are currently a rewriting of research results.

Re: We have rewritten the conclusion as your suggestion.

4. Authors stated that all Indel and SSR markers were separated using 4% polyacrylamide gel electrophoresis and silver stains for visualization. The electropherograms should be added into the Supplementary file.

Re: As your suggestion, we have added an electropherogram as Figure S1 (in line 150 and line 400), showing representative marker analyses on screening recombinants in the segregating population.

5. Extensive editing of English language is required.

Re: We have a native English speaker polish the manuscript. Also we have edited the paper extensively using Grammarly.

Reviewer 2 Report

Rice is one of the most important agricultural crops. Therefore, all studies related to increasing the yield of rice are relevant. In the presented manuscript, a large experiment was carried out, many types of rice were studied, so there is no doubt about the results obtained. Although more precise concrete evidence is required for the participation of the candidate gene in the process of rice growth. The manuscript is carefully designed, there are only minor remarks.

51- serine and cysteine proteases are different proteases and cannot be encoded by the same gene

Figure 2 f-h -"HET" is a marker? specify

Figure 4d,e,f - hard to see

There is no description of the results of table 1. What does the column "Environment E1,2,3" mean?

313- CO2

321 -9311?

323 -reference

Make references according to the rules of the journal

Author Response

1. 51- serine and cysteine proteases are different proteases and cannot be encoded by the same gene

Re: Thank you for your comments. We have corrected the error (in line 51) and cited a recent reference on the NAL1 gene.  

2. Figure 2 f-h -"HET" is a marker? Specify

Re: Thanks. We have added the notes for HET in the figure legend. HET represents the heterozygous genotype at the marker M9060.

3. Figure 4d,e,f - hard to see

Re: We have updated Figure 4 with a more clarity Figure 4d-f.

4. There is no description of the results of table 1. What does the column "Environment E1,2,3" mean?

Re: Thanks. We have added some results of Table 1 in lines 87-92. We also provide the footnote (in line 246) to explain what E1, E2, and E3 stand for. E1, E2, E3 represent three different conditions the CSSL population grown, respectively.

5. 313- CO2, 321 -9311,

Re: We have corrected the mistakes and checked throughout the manuscript again. 

6. 323 -reference

Re: We have added the reference (ref 37) for DNA extraction.

7. Make references according to the rules of the journal

Re: We have checked and re-ordered the cited references following the rules of journal.